# Ultra-Low Dose CT Chest in Acute COVID-19 Pneumonia: A Pilot Study from India

**DOI:** 10.3390/diagnostics13030351

**Published:** 2023-01-18

**Authors:** Mandeep Garg, Shritik Devkota, Nidhi Prabhakar, Uma Debi, Maninder Kaur, Inderpaul S. Sehgal, Sahajal Dhooria, Ashish Bhalla, Manavjit Singh Sandhu

**Affiliations:** 1Department of Radiodiagnosis & Imaging, Post Graduate Institute of Medical Education and Research (PGIMER), Chandigarh 160012, India; 2Department of Pulmonary Medicine, Post Graduate Institute of Medical Education and Research (PGIMER), Chandigarh 160012, India; 3Department of Internal Medicine, Post Graduate Institute of Medical Education and Research (PGIMER), Chandigarh 160012, India

**Keywords:** COVID-19, CT, radiation dose, low dose CT, ultra-low dose CT

## Abstract

The rapid increase in the number of CT acquisitions during the COVID-19 pandemic raised concerns about increased radiation exposure to patients and the resultant radiation-induced health risks. It prompted researchers to explore newer CT techniques like ultra-low dose CT (ULDCT), which could improve patient safety. Our aim was to study the utility of ultra-low dose CT (ULDCT) chest in the evaluation of acute COVID-19 pneumonia with standard-dose CT (SDCT) chest as a reference standard. This was a prospective study approved by the institutional review board. 60 RT-PCR positive COVID-19 patients with valid indication for CT chest underwent SDCT and ULDCT. ULDCT and SDCT were compared in terms of objective (noise and signal-to-noise ratio) and subjective (noise, sharpness, artifacts and diagnostic confidence) image quality, various imaging patterns of COVID-19, CT severity score and effective radiation dose. The sensitivity, specificity, positive and negative predictive value, and diagnostic accuracy of ULDCT for detecting lung lesions were calculated by taking SDCT as a reference standard. The mean age of subjects was 47.2 ± 10.7 years, with 66.67% being men. 90% of ULDCT scans showed no/minimal noise and sharp images, while 93.33% had image quality of high diagnostic confidence. The major imaging findings detected by SDCT were GGOs (90%), consolidation (76.67%), septal thickening (60%), linear opacities (33.33%), crazy-paving pattern (33.33%), nodules (30%), pleural thickening (30%), lymphadenopathy (30%) and pleural effusion (23.33%). Sensitivity, specificity and diagnostic accuracy of ULDCT for detecting most of the imaging patterns were 100% (*p* < 0.001); except for GGOs (sensitivity: 92.59%, specificity: 100%, diagnostic accuracy: 93.33%), consolidation (sensitivity: 100%, specificity: 71.43%, diagnostic accuracy: 93.33%) and linear opacity (sensitivity: 90.00%, specificity: 100%, diagnostic accuracy: 96.67%). CT severity score (range: 15–25) showed 100% concordance on SDCT and ULDCT, while effective radiation dose was 4.93 ± 1.11 mSv and 0.26 ± 0.024 mSv, respectively. A dose reduction of 94.38 ± 1.7% was achieved with ULDCT. Compared to SDCT, ULDCT chest yielded images of reasonable and comparable diagnostic quality with the advantage of significantly reduced radiation dose; thus, it can be a good alternative to SDCT in the evaluation of COVID-19 pneumonia.

## 1. Introduction

Computed tomography (CT) of the chest has played an important role in the radiological evaluation of COVID-19 pneumonia [1,2]. Real-time reverse transcription polymerase chain reaction (RT-PCR) is the gold standard test to confirm the diagnosis of COVID-19; however, its false negative rate and inability to assess the severity of disease and response to treatment of COVID-19 pneumonia led to growing utilization of CT chest [3]. Moreover, a significant number of COVID-19 patients continue to have lingering illnesses even in their post-recovery phase, and chest CT has proven to be a useful tool for monitoring the disease’s progress and its complications [4,5,6,7].

CT exposes patients to harmful ionizing radiation, and that has always been a matter of concern. But during the ongoing COVID-19 pandemic, the sudden increase in the number of CT examinations brought to the fore the issues related to patient safety, as the CT exposure of a large population (with many undergoing recurrent CT scans) in a short period is feared to have increased cumulative effective dose (CED) to the individuals [8]. CT radiation carries the risk of producing double-stranded DNA breaks and chromosomal aberrations that, in turn, can lead to genetic mutations and increased cancer risk in humans [9]. These are probabilistic, long-term carcinogenic effects of radiation which may sometimes take decades to manifest.

CT scanning is the major source of radiation exposure for patients from diagnostic medical imaging. The effective radiation dose delivered to the patient during a standard dose CT (SDCT) chest varies between 4–7 millisievert (mSv) [10,11]. Recent advancements in CT technology have helped in overcoming this issue with the introduction of low-dose CT (LDCT) and ultra-low dose CT (ULDCT), which exposes the patients to radiation in the range of 1–4 mSv, and <1 mSv, respectively [12]. Thus, the use of chest LDCT and ULDCT in the evaluation of patients with COVID-19 pneumonia can be a prudent way of curtailing radiation exposure and its health hazards [13].

Currently, LDCT and ULDCT are used mainly as screening tools for lung cancer screening and whole-body CT in multiple myeloma [14,15], as these techniques result in somewhat compromised image quality. With the technological advances in contemporary CT scanners, new research is underway to utilize LDCT and ULDCT more often with an aim to achieve diagnostic quality images at a much-reduced radiation dose. Our present study is also an endeavor in this direction to evaluate the diagnostic accuracy of ULDCT scans in the detection of various pulmonary parenchymal abnormalities of acute COVID-19 pneumonia, taking SDCT as the reference standard.

## 2. Materials and Methods

### 2.1. Study Design and Sample

This was a pilot study done on the Indian population, approved by the institutional review board (Institute Ethics Committee/2021/SPL-1074). A total of 60 consecutive patients were included prospectively in this study who were referred for CT chest from 29 May 2021 to 31 July 2021. As this was a prospective study without our populations’ previous data, a systematic computation of the sample size was not performed, and a viable, practicable and realistic sample size was planned. All the patients were COVID-19 RT-PCR-positive, with hypoxemia or respiratory distress or the ones not responding to standard treatment. Verbal informed consent was obtained from all the subjects, and written consent was obtained from patients’ attendants by proxy as per the prevailing guidelines of the hospital infection control committee at that time. Patients aged less than 18 years or those who did not consent to participate in the study were excluded. Clinical information, including the demographic profile, symptoms, clinical examination details, associated comorbidities, oxygen saturation levels and laboratory data, was collected from the medical records.

### 2.2. CT Acquisition and Protocol

SDCT and ULDCT scans were acquired successively on the included study participants. All scans were obtained in the dedicated COVID-19 CT suite on a 256-slice CT scanner (Philips BrillianceiCT256; Koninklijke Philips N.V., Netherlands). Scanning was performed in a supine position, and a helical dataset of images was acquired from lung apices to the domes of the diaphragm. HRCT images were reconstructed (using hybrid iterative reconstruction software iDose level 6) into 1 mm sections at 0.5 mm increment [reconstruction filter: Lung enhanced (L) for lung window and standard (B) for mediastinal window].

SDCT scans were acquired at 120 kVp with automatic exposure control (AEC) modulated tube current, while ULDCT scans were acquired at 80 kVp and 25 mAs with fixed tube current. The details of SDCT and ULDCT acquisition protocols are summarized in Table 1.

### 2.3. Image Interpretation and Analysis

Two experienced chest radiologists with 20 years (MG) and 10 years (UD) of experience evaluated the images who were blinded to the clinical data. Each of them first evaluated ULDCT scans independently, followed by SDCT scans, and recorded their findings for objective and subjective image quality, various imaging patterns of COVID-19 pneumonia, CT severity score and effective radiation dose. The data thus obtained from the analysis of ULDCT and SDCT scans were put to comparison, keeping SDCT as the reference standard.

Objective image quality assessment was done by obtaining image noise and signal-to-noise ratio (SNR). Region of interest (ROI) of size ~0.5 cm^2^ was placed in the tracheal lumen just before bifurcation without touching the tracheal wall. Image noise is the standard deviation of the attenuation, and SNR is the ratio of mean attenuation to the standard deviation of attenuation.

Subjective image quality was evaluated by subjective image noise, sharpness and diagnostic acceptability. Subjective image noise was scored on a 3-point scale ((1): minimum or no noise, (2): acceptable noise, (3): unacceptable noise). Sharpness was assessed on a 3-point scale ((1): sharp, (2): average, (3): blurry). Artifacts were assessed on a 3-point scale ((1): no artifacts, (2): artifacts are present but not affecting diagnostic confidence, (3): artifacts are present and affecting diagnostic confidence). Diagnostic confidence for accurately detecting imaging findings of COVID-19 pneumonia was also evaluated on a 3-point scale ((1): high diagnostic standard, (2): acceptable diagnostic standard, (3): poor diagnostic standard). The imaging findings that were assessed in ULDCT and SDCT included ground glass opacities (GGOs), consolidation, crazy paving, halo sign, septal thickening, linear opacity, air bronchogram, pleural thickening, nodules and bronchiectasis. Other than these main findings, the presence of pleural effusion, lymphadenopathy, pericardial effusion, cavitation, pneumothorax, pneumomediastinum and hydropneumothorax was also noted.

A semi-quantitative scoring system devised by Pan et al. was used for calculating the CT severity score of COVID-19 pneumonia. Each lobe of the lung was given a score of 0–5 based on the percentage of involvement: a score of 0 for 0% involvement, 1 for <5% involvement, 2 for 5–25% involvement, 3 for 26–50% involvement, 4 for 51–75% involvement and 5 for >75% involvement. The total score for both lungs (5 lobes) ranged from 0–25 [16].

### 2.4. Dose Calculation

Volume CT dose index (CTDI_vol_) and dose-length product (DLP) were obtained from the dose report. Effective radiation dose was calculated by multiplying DLP by the conversion factor (k) of 0.0144 for thoracic imaging as provided by International Commission on Radiological Protection (ICRP) 103 [17]. Further, the net effective radiation dose reduction (between ULDCT and SDCT) was expressed in percentage (%).

### 2.5. Statistical Analysis

Analysis was done with Statistical Package for Social Sciences (SPSS) software, IBM manufacturer, Chicago, USA, version 25.0. Quantitative variables were expressed as mean ± standard deviation. To compare quantitative variables, Student’s *t*-test was used. Qualitative variables were analyzed using the Chi-Square test and Fisher’s exact test. SDCT was used as the standard reference, and sensitivity, specificity, positive predictive value (PPV), negative predictive value (NPV) and diagnostic accuracy of ULDCT for the detection of lung parenchymal lesions related to COVID-19 were calculated with a 95% confidence interval. Interobserver variability was interpreted according to the classification for k (kappa) as follows:0–0.20, poor agreement;0.21–0.40, fair agreement;0.41–0.60, moderate agreement;0.61–0.80, substantial agreement; and0.81–1.00, almost perfect agreement.A *p*-value of less than 0.05 was considered statistically significant.

## 3. Results

Of the 71 subjects referred for chest CT during the study period, nine subjects refused to consent to participate, while two subjects were aged less than 18 years. Hence, a total of 60 patients were included in the study, as depicted in the enrolment flowchart in Figure 1.

The demographics, clinical features and laboratory findings, and the distribution of imaging patterns on chest CT are summarized in Table 2. The mean age of study participants was 47.2 ± 10.7 years, with two-thirds being male patients (n = 40, 66.67%). The mean duration between symptom to CT scan was 11.2 ± 2.2 days and between RT-PCR to CT scan was 8 ± 2.5 days.

The mean CTDIvol and DLP of SDCT were 9.45 ± 2.7 mGy and 352.57 ± 79.36 mGy.Cm, respectively. Mean CTDIvol and DLP of ULDCT were 0.5 mGy and 18.59 ± 1.72 mGy.Cm, respectively. The calculated mean effective radiation dose for SDCT and ULDCT were 4.93 ± 1.11 mSv and 0.26 ± 0.024 mSv, respectively. The net effective radiation reduction of ULDCT to that of SDCT scan was 94.38 ± 1.7%. The dose indices and image quality assessment are provided in Table 3.

The mean SNR of SDCT and ULDCT were 31.35 ± 3.32 and 14.53 ± 1.55. Of the 60 scans included in the study, all of the SDCT scans had no or minimal noise, sharp image quality and high diagnostic confidence. Only 10% of the SDCT scans showed artifacts without affecting the diagnostic confidence. ULDCT scans showed no or minimal noise in 90% of the scans, acceptable noise in 6.67%, while 3.33% had unacceptable levels of noise. Ninety percent of the ULDCT scans yielded sharp images, while 10% were of average sharpness. There were artifacts in 13.33% of the ULDCT scans; however, they did not affect the diagnostic confidence. ULDCT scans had image quality of high diagnostic confidence in 93.33% of the images.

Figure 2 depicts the imaging findings as detected by SDCT and ULDCT scans in the bar diagram. Common imaging findings observed on SDCT were GGOs, consolidation, septal thickening, linear opacities, crazy paving pattern, air bronchogram and halo sign which were present in 54 (90%), 46 (76.67%), 36 (60%), 20 (33.33%), 20 (33.33%), 10 (16.67%) and 10 (16.67%) patients, respectively. Nodules were seen in 18 (30%) patients, bronchiectasis in eight (13.33%) patients, pleural thickening in 18 (30%) patients, pleural effusion in 14 (23.33%) patients and lymphadenopathy in 18 (30%) patients. Two (3.33%) patients each had pericardial effusion and tree-in-bud pattern, while eight (13.33%) patients showed cavitary changes. Pneumothorax was seen in four (6.67%), while pneumomediastinum and hydropneumothorax were present in six (10%) patients each.

Table 4 shows the sensitivity, specificity, PPV, NPV and diagnostic accuracy of ULDCT in detecting the imaging patterns of COVID-19 pneumonia with SDCT as the standard reference. The diagnostic accuracy of ULDCT for the detection of GGOs, consolidation, and linear opacity was 93.33% [(83.80–98.15%), *p* < 0.001], 93.33% [(83.80–98.15%), *p* < 0.001] and 96.67% [(88.47–99.59%), *p* < 0.001], respectively; while, diagnostic accuracy for detection of rest of the imaging patterns was 100.00% [(94.04–100.00%), *p* < 0.001] (Figure 3, Figure 4 and Figure 5). There was almost perfect interobserver agreement (k ~0.82–1). The semi-quantitative CT severity score was in the range of 15–25, and it showed 100% concordance between scores obtained on SDCT and ULDCT.

## 4. Discussion

CT chest plays a pivotal role in the evaluation of COVID-19 pneumonia [1,2,18,19]. The humongous number of patients affected by COVID-19 worldwide during the current pandemic increased the demand for chest CT with some of the patients undergoing frequent repeat scanning, which can result in increasing patients’ CED [8,20,21]. This prompted the healthcare providers to explore alternate imaging modalities which don’t expose the patients to ionizing radiation, like ultrasound and MRI [22,23].

However, CT remains an indispensable imaging tool in COVID-19; thus, it becomes imperative to explore new CT techniques like LDCT and ULDCT, which can cut down the radiation exposure to the patient while being able to provide quality diagnostic images. Our prospective comparative study of ULDCT with SDCT scan to evaluate the imaging findings of COVID-19 pneumonia showed comparable image quality and sensitivity, specificity and diagnostic accuracy between these two chest CT techniques, as detailed in the results and Table 2, Table 3 and Table 4.

GGOs were the commonest imaging pattern of COVID-19 pneumonia seen in our study in 90% of patients, which is in concordance with the results of a systematic review and meta-analysis done by Garg et al. [16]. In the current study, when compared to SDCT, ULDCT could identify GGOs in 93.33% (83.80–98.15%) of subjects with a sensitivity of 92.59% (82.11–97.94%) and a specificity of 100.00% (54.07–100.00%). In another prospective study done by Zarei et al., the sensitivity and specificity for the detection of GGOs on ULDCT scans were much lower at 62% and 66%, respectively [24]. The most plausible explanation for this could be that the majority of participants in our study group had a more extensive and diffuse distribution of GGOs, and that could have led to its higher pick-up rate. The sensitivity of ULDCT for identification of consolidation was found to be 100% (92.29–100.00%), while the specificity was much lower at 71.43% (41.90–91.61%), which is due to an erroneous interpretation of GGOs as consolidations on ULDCT in two patients in our study cohort.

The reported sensitivity of linear opacities on ULDCT was 90% (68.30–98.77%), with a specificity of 100.00% (91.19–100.00%). Linear opacities were missed on two ULDCT scans. The rest of the major abnormal findings, including crazy paving, halo sign, air-bronchogram and pulmonary nodules, showed 100% sensitivity and 100% specificity. Each imaging pattern witnessed on SDCT was detected by ULDCT with high sensitivity, specificity, PPV, NPV and diagnostic accuracy. Our observations are in agreement with Greffier et al., who, in a similar study, compared the diagnostic value of ULDCT with SDCT and found comparable sensitivity and specificity of ULDCT (98.9% and 99% respectively) to SDCT [25].

We found ULDCT scans had reduced objective image quality with mean SNR of 14.53 ± 1.55, which is about 46.34% of the SNR of SDCT scans. However, the subjective image quality parameters of ULDCT were in keeping with SDCT scans, with only two ULDCT scans showing unacceptable levels of noise. Nevertheless, the overall diagnostic quality of ULDCT chest images was acceptable for interpretation. This is again in concordance with the study by Samir et al., who also reported slight derangement in the image quality with ULDCT, but without any impairment in diagnostic confidence [26]. Recently, a few more authors have also reported similar results and diagnostic accuracy of ULDCT for the identification of COVID-19-related pulmonary findings as compared to SDCT [24,25,26,27,28,29].

In a prospective study, Samir et al. reported mean CTDI and DLP of 1.1 ± 0.3 mGy and 42.2 ± 7.9 mGy.cm, respectively, for ULDCT with mean effective radiation of 0.59 mSv [26]. Their ULDCT acquisition was done at 120 kVp and 30 mAs. We did ULDCT at 80 kVp and 25 mAs and reported mean CTDI and DLP values of 0.5 mGy and 18.59 ± 1.72 mGy.Cm, respectively, with a mean effective radiation dose of 0.26 ± 0.024 mSv and could achieve a net effective dose reduction of 94.38 ± 1.7%. In another study by Zarei et al., ULDCT scans were acquired at 80 kVp and 25 mAs with a mean effective radiation dose of 0.246 ± 0.055 mSv [24]. These results are comparable to what we have also reported in the present study.

The ionizing radiation (including that from CT scans) carries potential health hazards in the form of inducing heritable genetic damage and increased cancer risk, which increases proportionately with the increase in radiation dose [30,31]. Thus, in an attempt to enhance patient safety, researchers are evaluating newer CT protocols and techniques like LDCT and ULDCT, which aim at minimizing patients’ radiation exposure. In one such prospective study on 209 patients, Sakane H et al. found that there was no change in the number of double-strand breaks and chromosomal error prior to and after LDCT, concluding that LDCT did not have any effect on the human DNA, while double-strand breaks and aberrations were noted in the DNA of subjects who underwent SDCT chest [32]. The radiation dose delivered to the patients with ULDCT is further less than LDCT, equivalent only to a couple of chest x-rays, thus minimizing the radiation hazard to nearly unrecognizable. But, ULDCT is much better than chest X-ray and offers many advantages as it provides high resolution, multi-planar three-dimensional images and its diagnostic accuracy is comparable to SDCT [26,27,28,33].

There are a few limitations in our study. One, it is a single-center study with a small sample size. Two, our study cohort has patients with acute COVID-19 pneumonia with respiratory distress, and the majority of these patients had moderate-to-severe grades of infection with significant lung involvement. So, we could not evaluate and compare the subtle lung abnormalities of the milder form, and it could have given some bias to our results. And lastly, we used fixed kVp and mAs for ULDCT scans and didn’t take into account the weight/BMI of the subject participants, which is an important parameter while selecting the CT acquisition parameters as it can affect the image quality.

## 5. Conclusions

ULDCT scan of the chest with properly devised CT acquisition parameters can yield images of reasonable and comparable diagnostic quality to SDCT with the advantage of significantly reduced radiation dose to the patients. The feasibility of ULDCT lies in the ease of its doing by simply changing the acquisition parameters without the need for any additional equipment or software. It can be utilized more frequently for imaging in acute COVID-19 pneumonia and even in the post-recovery phase as a follow-up tool. However, further larger, multicentric studies are needed to validate its diagnostic accuracy in the detection of COVID-19-related pulmonary parenchymal abnormalities.

## Figures and Tables

**Figure 1 diagnostics-13-00351-f001:**
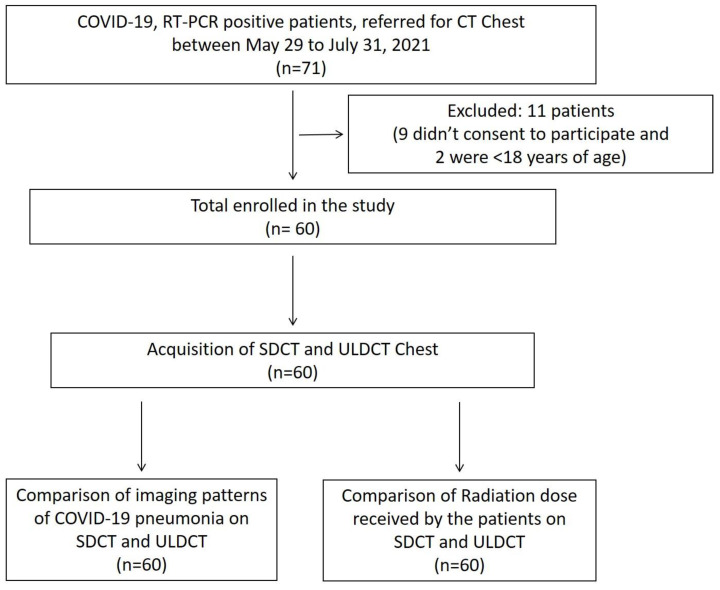
Flow chart depicting the inclusion process and study design.

**Figure 2 diagnostics-13-00351-f002:**
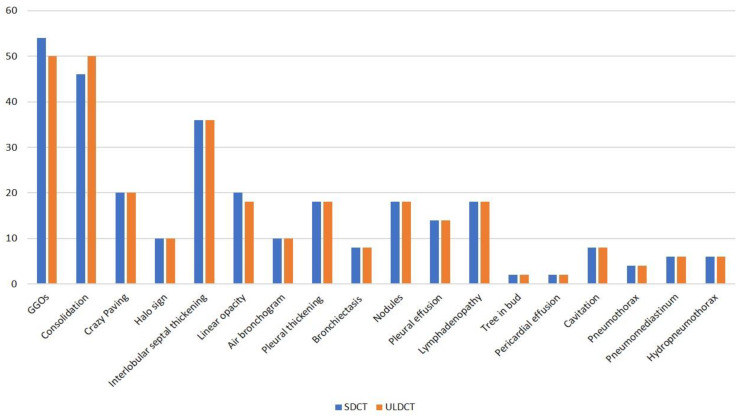
Bar diagram showing various imaging patterns of COVID-19 pneumonia and its frequency in the study cohort, as detected by standard dose CT (SDCT) and ultra-low-dose CT (ULDCT).

**Figure 3 diagnostics-13-00351-f003:**
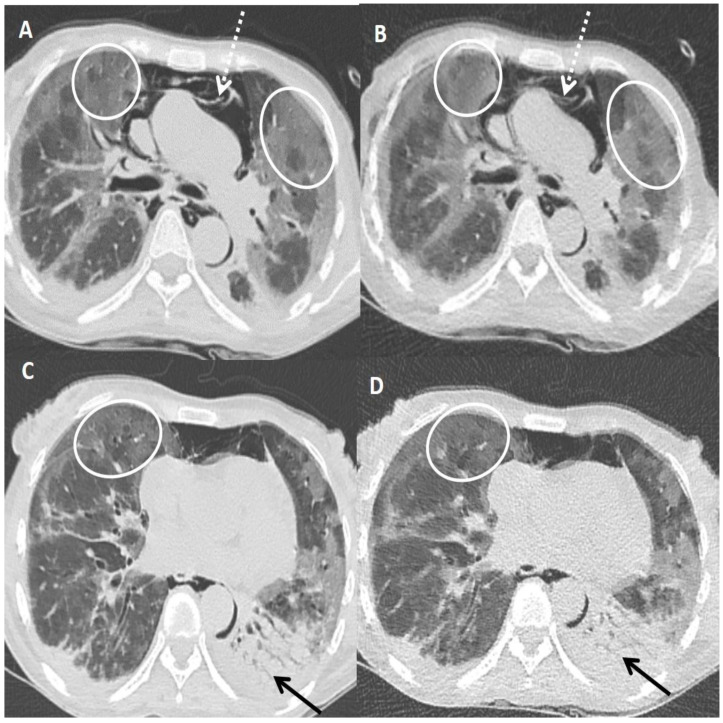
A 45-years old female patient tested positive for COVID-19. Standard dose CT (SDCT) chest (**A**,**C**) and corresponding ultra-low dose CT (ULDCT) chest (**B**,**D**) showing areas of GGOs (white circles) and patchy consolidation with air bronchogram (black arrows). Pneumomediastinum was also seen (dotted white arrows). The effective radiation dose for SDCT and ULDCT was 4.13 mSv and 0.25 mSv, respectively, while the CT severity score calculated on both SDCT and ULDCT was similar (19/25).

**Figure 4 diagnostics-13-00351-f004:**
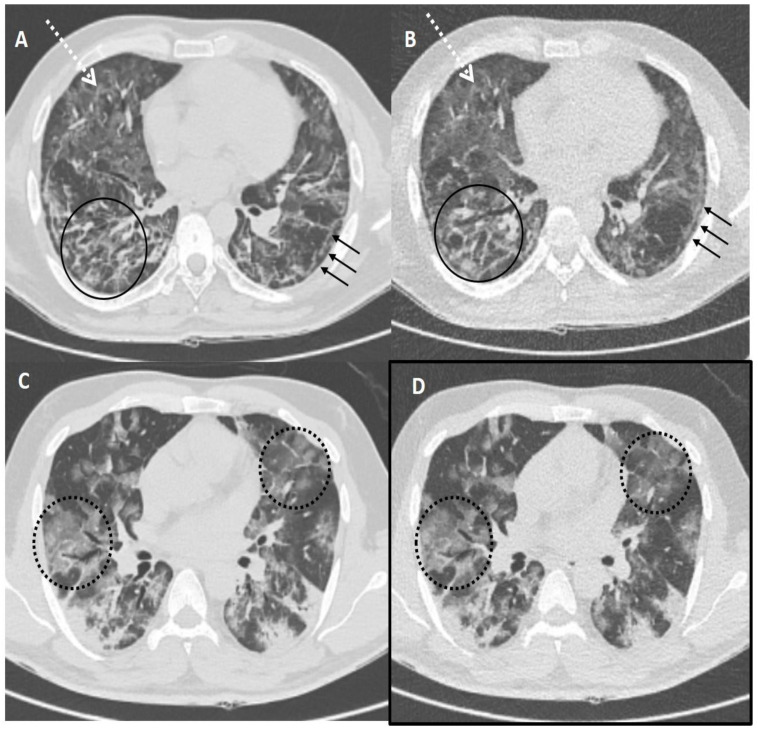
Comparison of standard-dose CT (SDCT) chest and ultra-low dose CT (ULDCT) chest in two different COVID-19 patients: (**A**,**B**) A 51-years old male with SDCT (**A**) and corresponding ULDCT (**B**) images showing interlobular septal thickening/reticulation (black circles) and parenchymal bands (black arrows), with interspersed areas of GGOs in both lungs (dotted white arrows). (**C**,**D**) A 45-years old male patient with SDCT chest (**C**) and corresponding ULDCT chest (**D**) showing areas of GGOs with interlobular septal thickening giving a crazy-paving pattern (dotted circles). The CT severity score calculated on SDCT and ULDCT was found to be similar in both patients, while the effective radiation dose for SDCT was 4.63 mSv and 4.81 mSv, and for ULDCT was 0.26 mSv and 0.28 mSv, respectively.

**Figure 5 diagnostics-13-00351-f005:**
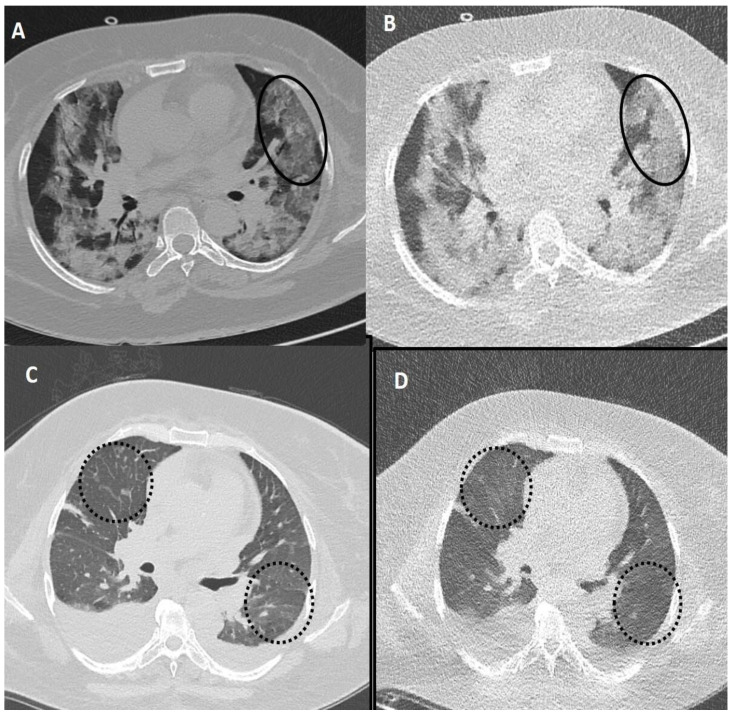
Fallacies of ultra-low dose CT (ULDCT) chest in two different COVID-19 patients: (**A**,**B**) A 53-years old female patient with SDCT (**A**) showing areas of GGOs in the left lung (black circle) and in the corresponding ULDCT (**B**) these areas of GGO’s were misinterpreted as consolidation (black circle). (**C**,**D**) SDCT images in another patient (**C**) showing subtle interlobular septal thickening in bilateral lungs (dotted circles) that was missed on the corresponding ULDCT (**D**) images (dotted circles). CT severity score calculated on SDCT and ULDCT, however, was similar in both patients.

**Table 1 diagnostics-13-00351-t001:** Detailed acquisition parameters of SDCT and ULDCT chest.

Parameters	SDCT Chest	ULDCT Chest
Scanning mode	Helical	Helical
Tube potential (kV)	120	80
Tube current time (mAs)	AEC modulated	25
Tube current modulation technique	AEC	Fixed tube current
Pitch	0.758	0.758
Rotation time (s)	0.5	0.5
Slice thickness (in mm)	10	10
Iterative reconstruction technique	Hybrid iterative reconstruction software iDose level 6	Hybrid iterative reconstruction software iDose level 6
Orientation	Head first	Head first

Abbreviations: SDCT = Standard dose computed tomography, ULDCT = Ultra low dose computed tomography, AEC = Automated exposure control.

**Table 2 diagnostics-13-00351-t002:** Demographic, clinical, laboratory details and distribution of lung involvement on chest CT of the study subjects.

Patient Characteristics	Observations (n = 60)
**Demographics**	**Frequency**
Age (years)	47.2 ± 10.7 (mean ± S.D)
**Gender**	**Frequency**
Male	40 (66.67%)
Female	20 (33.33%)
**Clinical features**	**Frequency**
Fever	54 (90.00%)
Cough	58 (96.67%)
Dyspnea	48 (80.00%)
Myalgia	30 (50.00%)
Fatigue	30 (50.00%)
Anosmia or ageusia	38 (63.33%)
Sore throat	30 (50.00%)
**Clinical history**	**Frequency**
Diabetes	20 (33.33%)
Hypertension	18 (30.00%)
CKD	6 (5.00%)
Coronary artery disease	4 (6.67%)
COPD	2 (3.33%)
Past history of tuberculosis	5 (8.33%)
Smoker	14 (23.33%)
Alcoholic	15 (25.00%)
**Laboratory data**	**Frequency**
Anaemia	9 (15.00%)
Leucocytosis	10 (16.67%)
Thrombocytopenia	7 (11.67%)
Deranged RFT	6 (5.00%)
Raised LDH	40 (66.67%)
Raised CRP	42 (70.00%)
Raised procalcitonin	30 (50.00%)
Raised ferritin	34 (56.66%)
Raised troponin T	24 (40.00%)
Raised d-dimer	32 (53.33%)
**Duration from:**	**Days**
Symptom to CT scan	11.2 ± 2.2 (mean ± S.D)
RT-PCR to CT scan	8 ± 2.5 (mean ± S.D)
**Distribution of lung abnormalities**	**Frequency**
Bilateral	60 (100%)
Peripheral	38 (63.33%)
Diffuse	18 (30.00%)
Random	4 (6.67%)
RUL	46 (76.67%)
RML	52 (86.67%)
RLL	60 (100%)
LUL	46 (76.67%)
LLL	54 (90.00%)

Abbreviations: SDCT = Standard dose computed tomography, ULDCT = Ultra low dose computed tomography, S.D = Standard deviation, CKD = Chronic kidney disease, COPD = Chronic obstructive pulmonary disease, WBC = White blood cell count, RFT = Renal function test, LDH = lactate dehydrogenase, CRP = C-reactive protein, RT-PCR- Real time-reverse transcription polymerase chain reaction, RUL = Right upper lobe, RML = Right middle lobe, RLL = Right lower lobe, LUL = Left upper lobe, LLL = Left lower lobe.

**Table 3 diagnostics-13-00351-t003:** Comparison of dose indices and image quality between SDCT and ULDCT.

Dose Indices and Image Quality		SDCT	ULDCT
CTDI_vol_ (mGy)		9.45 ± 2.70 (mean ± S.D)	0.5 ± 00 (mean ± S.D)
DLP (mGycm)		352.57 ± 79.36 (mean ± S.D)	18.59 ± 1.72 (mean ± S.D)
Effective radiation dose (mSv)		4.93 ± 1.11 (mean ± S.D)	0.26 ± 0.02 (mean ± S.D)
Net effective radiation dose reduction		94.38 ± 1.7% (mean ± S.D)	
SNR		31.35 ± 3.32 (mean ± S.D)	14.53 ± 1.55 (mean ± S.D)
Noise	No or minimum	60 (100%)	54 (90%)
	Acceptable	0%	4 (6.67%)
	Unacceptable	0%	2 (3.33%)
Sharpness	Sharp	60 (100%)	54 (90%)
	Average	0%	6 (10%)
	Blurry	0%	0%
Artifact	Absent	54 (90%)	52 (86.67%)
	Present but not affecting diagnostic confidence	6 (10%)	8 (13.33%)
	Present and affecting diagnostic confidence	0%	0%
Diagnostic confidence	High	60 (100%)	56 (93.33%)
	Acceptable	0%	4 (6.67%)
	Poor	0%	0%

Abbreviations: SDCT = Standard dose computed tomography, ULDCT = Ultra low dose computed tomography, CTDI_vol_ = Volume CT dose index, DLP = dose-length product, S.D = standard deviation, SNR = Signal to noise ratio.

**Table 4 diagnostics-13-00351-t004:** The performance of ULDCT in detecting the imaging patterns of COVID-19 pneumonia (with SDCT as the reference standard).

Imaging Patterns	TP	TN	FP	FN	Sensitivity	Specificity	PPV	NPV	Diagnostic Accuracy	Kappa	*p*-Value
GGOs	50	6	0	4	92.59% (82.11–97.94%)	100.00% (54.07–100.00%)	100.00% (88.1–100%)	60.00% (36.88–79.39%)	93.33% (83.80–98.15%)	0.82	<0.001
Consolidation	46	10	4	0	100.00% (92.29–100.00%)	71.43% (41.90–91.61%)	92.00% (83.40–96.34%)	100.00% (78.14–100%)	93.33% (83.80–98.15%)	0.82	<0.001
Crazy paving	20	40	0	0	100.00% (83.16–100.00%)	100.00% (91.19–100.00%)	100.00% (83.16–100.00%)	100.00% (91.19–100.00%)	100.00% (94.04–100.00%)	1	<0.001
Halo sign	10	50	0	0	100.00% (69.15–100.00%)	100.00% (92.89–100.00%)	100.00% (69.15–100.00%)	100.00% (92.89–100.00%)	100.00% (94.04–100.00%)	1	<0.001
Septal thickening/reticulation	36	24	0	0	100.00% (90.26–100.00%)	100.00% (85.75–100.00%)	100.00% (90.26–100.00%)	100.00% (85.75–100.00%)	100.00% (94.04–100.00%)	1	<0.001
Linear opacity	18	40	0	2	90.00% (68.30–98.77%)	100.00% (91.19–100.00%)	100.00% (76.2–100%)	95.23% (84.30–98.68%)	96.67% (88.47–99.59%)	0.95	<0.001
Air bronchogram	10	50	0	0	100.00% (69.15–100.00%)	100.00% (92.89–100.00%)	100.00% (69.15–100.00%)	100.00% (92.89–100.00%)	100.00% (94.04–100.00%)	1	<0.001
Pleural thickening	18	42	0	0	100.00% (81.47–100.00%)	100.00% (91.59–100.00%)	100.00% (81.47–100.00%)	100.00% (91.59–100.00%)	100.00% (94.04–100.00%)	1	<0.001
Bronchiectasis	8	52	0	0	100.00% (63.06–100.00%)	100.00% (93.15–100.00%)	100.00% (63.06–100.00%)	100.00% (93.15–100.00%)	100.00% (94.04–100.00%)	1	<0.001
Nodules	18	42	0	0	100.00% (81.47–100.00%)	100.00% (91.59–100.00%)	100.00% (81.47–100.00%)	100.00% (91.59–100.00%)	100.00% (94.04–100.00%)	1	<0.001
Pleural effusion	14	46	0	0	100.00% (76.84–100.00%)	100.00% (92.29–100.00%)	100.00% (76.84–100.00%)	100.00% (92.29–100.00%)	100.00% (94.04–100.00%)	1	<0.001
Lymphadenopathy	18	42	0	0	100.00% (81.47–100.00%)	100.00% (91.59–100.00%)	100.00% (81.47–100.00%)	100.00% (91.59–100.00%)	100.00% (94.04–100.00%)	1	<0.001
Tree-in-bud	2	58	0	0	100.00% (15.81–100.00%)	100.00% (93.84–100.00%)	100.00% (15.81–100.00%)	100.00% (93.84–100.00%)	100.00% (94.04–100.00%)	1	<0.001
Pericardial effusion	2	58	0	0	100.00% (15.81–100.00%)	100.00% (93.84–100.00%)	100.00% (15.81–100.00%)	100.00% (93.84–100.00%)	100.00% (94.04–100.00%)	1	<0.001
Cavitation	8	52	0	0	100.00% (63.06–100.00%)	100.00% (93.15–100.00%)	100.00% (63.06–100.00%)	100.00% (93.15–100.00%)	100.00% (94.04–100.00%)	1	<0.001
Pneumothorax	4	56	0	0	100.00% (39.76–100.00%)	100.00% (93.62–100.00%)	100.00% (39.76–100.00%)	100.00% (93.62–100.00%)	100.00% (94.04–100.00%)	1	<0.001
Pneumomediastinum	6	54	0	0	100.00% (54.07–100.00%)	100.00% (93.40–100.00%)	100.00% (54.07–100.00%)	100.00% (93.40–100.00%)	100.00% (94.04–100.00%)	1	<0.001
Hydropneumothorax	6	54	0	0	100.00% (54.07–100.00%)	100.00% (93.40–100.00%)	100.00% (54.07–100.00%)	100.00% (93.40–100.00%)	100.00% (94.04–100.00%)	1	<0.001

Abbreviations: TP = True positive, TN = True negative, FP = False positive, FN = False negative, PPV = Positive predictive value, NPV = Negative predictive value, SDCT = Standard dose computed tomography, ULDCT = Ultra low dose computed tomography, GGOs = Ground glass opacities.

## Data Availability

Not applicable.

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
