# Peer review of "Ultra-Low Dose CT Chest in Acute COVID-19 Pneumonia: A Pilot Study from India"

_diagnostics, 2023, doi:10.3390/diagnostics13030351_

Round 1

Reviewer 1 Report

The article is interesting, well written and of interest to radiological community.

The authors are using through whole text: 'radiation dose exposure, effective radiation dose, radiation dose, effective radiation. This should be reviewed and correct terms used. Maybe, this article could help: Homayounieh F, Holmberg O, Umairi RA, Aly S, Basevičius A, Costa PR, Darweesh A, Gershan V, Ilves P, Kostova-Lefterova D, Renha SK, Mohseni I, Rampado O, Rotaru N, Shirazu I, Sinitsyn V, Turk T, Van Ngoc Ty C, Kalra MK, Vassileva J. Variations in CT Utilization, Protocols, and Radiation Doses in COVID-19 Pneumonia: Results from 28 Countries in the IAEA Study. Radiology. 2021 Mar;298(3):E141-E151. doi: 10.1148/radiol.2020203453. Epub 2020 Nov 10. PMID: 33170104; PMCID: PMC7673104.

Why not use only dose indices in comparison? 'Effective radiation dose' is approximation and directly proportional to the CTDI (E=0.0144CTDI).

How can CT increase background radiation?

Whole para before Table 4 is data that are going to be repeated in the Table 4. Data should be presented only once and it should be omitted from the para.

In discussion you declare that the results are 'almost similar'? It could be that results are similar, comparable, close, almost the same? Not sure what 'almost similar means'.

Author Response

RESPONSE TO COMMENTS OF REVIEWER 1

Point 1: The authors are using through whole text: 'radiation dose exposure, effective radiation dose, radiation dose, effective radiation. This should be reviewed and correct terms used. Maybe, this article could help: Homayounieh F, Holmberg O, Umairi RA, Aly S, Basevičius A, Costa PR, Darweesh A, Gershan V, Ilves P, Kostova-Lefterova D, Renha SK, Mohseni I, Rampado O, Rotaru N, Shirazu I, Sinitsyn V, Turk T, Van Ngoc Ty C, Kalra MK, Vassileva J. Variations in CT Utilization, Protocols, and Radiation Doses in COVID-19 Pneumonia: Results from 28 Countries in the IAEA Study. Radiology. 2021 Mar;298(3):E141-E151. doi: 10.1148/radiol.2020203453. Epub 2020 Nov 10. PMID: 33170104; PMCID: PMC7673104.

Response 1: We thank the reviewer for this observation. We have rectified this and the terms ‘radiation dose exposure’ and ‘effective radiation’ have been appropriately replaced throughout the text with ‘radiation dose’ and ‘effective radiation dose”. We have also added the reference suggested by the reviewer and one more latest reference at number 20 & 21 in the manuscript as mentioned below:

  1. Homayounieh F, Holmberg O, Umairi RA, et al. Variations in CT Utilization, Protocols, and Radiation Doses in COVID-19 Pneumonia: Results from 28 Countries in the IAEA Study. Radiology. 2021;298(3):E141-E151. doi:10.1148/radiol.2020203453

  1. Garg, M.; Karami, V.; Moazen, J. et al. Radiation Exposure and Lifetime Attributable Risk of Cancer Incidence and Mortality from Low- and Standard-Dose CT Chest: Implications for COVID-19 Pneumonia Subjects. Diagnostics202212, 3043. https://doi.org/10.3390/diagnostics12123043

Point 2: Why not use only dose indices in comparison? 'Effective radiation dose' is approximation and directly proportional to the CTDI (E=0.0144CTDI).

Response 2: We thank reviewer for his valuable comment and we agree that effective radiation dose is an approximate value derived from DLP. We have given CTDIvol and DLP values in table 3, however, in addition to these dose indices, we have also calculated ‘effective radiation dose’ and used it in our manuscript discussion. ‘Effective Radiation dose’ - is an easier way of interpreting the radiation dose related exposure, not only for the radiologists but also for the referring/treating physicians and the patients. Moreover, ‘effective radiation dose’ is also used frequently while calculating the ‘cumulative radiation dose’ in patients; and in various radiation safety limits/guidelines.

Point 3: How can CT increase background radiation?

Response 3: We thank the reviewer for pointing this out. We have removed this line both from introduction and discussion in the revised text.

Point 4: Whole para before Table 4 is data that are going to be repeated in the Table 4. Data should be presented only once and it should be omitted from the para.

Response 4: We thank reviewer for this observation. We have truncated the paragraph before Table 4 and the repetition has been deleted.

Point 5: In discussion you declare that the results are 'almost similar'? It could be that results are similar, comparable, close, almost the same? Not sure what 'almost similar means'.

Response 5: We agree with reviewer’s observation. We have rectified it as ‘comparable’

Reviewer 2 Report

In this prospective study, authors investigated the diagnostic performances of ULDCT, using SDCT as reference standard. The main limitation of the study regards the choice of the included population, i.e. only severe COVID-19 patients that usually have a major pulmonary involvement. It would be interesting to evaluate the performances of ULDCT in mild COVID-19 patients, mainly regarding the detection of GGOs that are the most common CT findings.

-       The title should be modified to also include the term “severe”, considering that only patients with a high CT score have been included in the study

-       Did authors apply additional inclusion/exclusion criteria (e.g. maximum time interval from the onset of symptoms or RT-PCR and CT scan, exclusion of low-quality images for respiratory artifacts)?

-       Table 3: the same number of decimals should be used for CTDI vol, DLP and Effective radiation dose

Author Response

RESPONSE TO COMMENTS OF REVIEWER 2

Point 1: In this prospective study, authors investigated the diagnostic performances of ULDCT, using SDCT as reference standard. The main limitation of the study regards the choice of the included population, i.e. only severe COVID-19 patients that usually have a major pulmonary involvement. It would be interesting to evaluate the performances of ULDCT in mild COVID-19 patients, mainly regarding the detection of GGOs that are the most common CT findings.

Response 1: We agree with the reviewer that patients in our cohort were having moderate to severe disease with significant pulmonary involvement. The lung parenchymal changes in patients with mild disease could not be evaluated in our study population and we have highlighted the same in the ‘limitations’ of the study.

Point 2: The title should be modified to also include the term “severe”, considering that only patients with a high CT score have been included in the study

Response 2: We thank the reviewer for his comment. We have not used the term ‘severe’ in our title, as this was not our inclusion criteria. Our inclusion criteria were RT-PCR positive, hospitalised patients with hypoxemia, or respiratory distress, or those who were not responding to standard treatment. The severity and extent of the lung involvement was known only after the CT scan was acquired; and the results of all patients showing high CT score were purely incidental.

Further, based on the CT severity score, not all patients of our cohort fell in ‘severe’ disease category. The disease is considered mild for CT severity score of <7, moderate for CT score of 8-17 and severe if it is >18 (references given below). We had 13 patients in the ‘moderate’ category (6 patients with CT severity score of 15, 2 with 16 and 5 having CT score of 17).

  1. Pan F, Ye T, Sun P, et al. Time Course of Lung Changes at Chest CT during Recovery from Gurumurthy B, Das S, Shetty S, et al. CT severity score: an imaging biomarker to estimate the severity of COVID-19 pneumonia in vaccinated and non-vaccinated population The Egyptian Journal of Radiology and Nuclear Medicine. 2022 Jan;53(1). PMCID: PMC9003164.
  2. Coronavirus Disease 2019 (COVID-19). Radiology. 2020;295(3):715-721. doi:10.1148/radiol.2020200370
  3. Saeed GA, Gaba W, Shah A, et al. Correlation between Chest CT Severity Scores and the Clinical Parameters of Adult Patients with COVID-19 Pneumonia. Radiol Res Pract. 2021;2021:6697677. Published 2021 Jan 6. doi:10.1155/2021/6697677

Point 3: Did authors apply additional inclusion/exclusion criteria (e.g. maximum time interval from the onset of symptoms or RT-PCR and CT scan, exclusion of low-quality images for respiratory artifacts)?

Response 3: No additional inclusion/exclusion criteria was used apart from the one written in the manuscript. No time interval was kept between scan and onset of symptoms. No scans were excluded from the study based on the quality of images/artifacts.

Point 4: Table 3: the same number of decimals should be used for CTDI vol, DLP and Effective radiation dose

Response 4: We thank the reviewer for this observation. We have corrected the same. Now only up to 2 decimals have been used for expressing the indices.
